# Endotoxin Tolerance in Abdominal Aortic Aneurysm Macrophages, In Vitro: A Case–Control Study

**DOI:** 10.3390/antiox9090896

**Published:** 2020-09-21

**Authors:** Lara T. Meital, Mark T. Windsor, Alesiya E. Maynard, Karl Schulze, Rebecca Magee, Jill O’Donnell, Pankaj Jha, Chaim Y. Meital, Maria Perissiou, Steven Coverdale, Jonathan Golledge, Anna V. Kuballa, Tom G. Bailey, Christopher D. Askew, Fraser D. Russell

**Affiliations:** 1Centre for Genetics, Ecology & Physiology, School of Health and Sport Sciences, University of the Sunshine Coast, Maroochydore, QLD 4558, Australia; Lara.meital@research.usc.edu.au (L.T.M.); Mark.Windsor@health.qld.gov.au (M.T.W.); alesiyamaynard@gmail.com (A.E.M.); maria.perissiou@port.ac.uk (M.P.); s.coverdale@griffith.edu.au (S.C.); akuballa@usc.edu.au (A.V.K.); Tom.Bailey@uq.edu.au (T.G.B.); CAskew@usc.edu.au (C.D.A.); 2Sunshine Vascular, Buderim, QLD 4556, Australia; Karl.svi@bigpond.com; 3Department of Surgery, Sunshine Coast University Hospital, Birtinya, QLD 4575, Australia; r.magee@ausdoctors.net (R.M.); Jill.ODonnell@health.qld.gov.au (J.O.); Pankaj.Jha@health.qld.gov.au (P.J.); 4Moffat Beach Family Medical Practice, Moffat Beach, QLD 4551, Australia; cmeital@bigpond.com; 5School of Medicine, Griffith University, Birtinya, QLD 4575, Australia; 6Queensland Research Centre for Peripheral Vascular Disease, College of Medicine and Dentistry, James Cook University, Townsville 4811, Australia; jonathan.golledge@jcu.edu.au; 7Department of Vascular and Endovascular Surgery, Townsville Hospital, Townsville 4810, Australia; 8Centre for Research on Exercise, Physical Activity and Health, School of Human Movement and Nutrition Sciences, The University of Queensland, St. Lucia, QLD 4072, Australia

**Keywords:** abdominal aortic aneurysm, endotoxin tolerance, lipid rafts, macrophages, toll-like receptor 4 (TLR4)

## Abstract

Macrophages are implicated in the pathogenesis of abdominal aortic aneurysm (AAA). This study examined the environmentally conditioned responses of AAA macrophages to inflammatory stimuli. Plasma- and blood-derived monocytes were separated from the whole blood of patients with AAA (30–45 mm diameter; *n* = 33) and sex-matched control participants (*n* = 44). Increased concentrations of pro-inflammatory and pro-oxidant biomarkers were detected in the plasma of AAA patients, consistent with systemic inflammation and oxidative stress. However, in monocyte-derived macrophages, a suppressed cytokine response was observed in AAA compared to the control following stimulation with lipopolysaccharide (LPS) (tumor necrosis factor alpha (TNF-α) 26.9 ± 3.3 vs. 15.5 ± 3.2 ng/mL, *p* < 0.05; IL-6 3.2 ± 0.6 vs. 1.4 ± 0.3 ng/mL, *p* < 0.01). LPS-stimulated production of 8-isoprostane, a biomarker of oxidative stress, was also markedly lower in AAA compared to control participants. These findings are consistent with developed tolerance in human AAA macrophages. As Toll-like receptor 4 (TLR4) has been implicated in tolerance, macrophages were examined for changes in TLR4 expression and distribution. Although TLR4 mRNA and protein expression were unaltered in AAA, cytosolic internalization of receptors and lipid rafts was found. These findings suggest the inflamed, pro-oxidant AAA microenvironment favors macrophages with an endotoxin-tolerant-like phenotype characterized by a diminished capacity to produce pro-inflammatory mediators that enhance the immune response.

## 1. Introduction

Abdominal aortic aneurysm (AAA), characterized by a complex, multifactorial pathogenesis, is a clinically silent vascular disease typically localized to the terminal aortic segment [1]. Degenerative by nature, the condition is associated with chronic inflammatory cell infiltration and destructive remodeling of aortic connective tissue that culminates in a focal loss of vessel wall integrity and full thickness dilation of the abdominal aorta [2,3]. AAA initiation and progression is strongly associated with aortic macrophage accumulation [4]. Macrophages, derived from circulating monocytes or monocyte reservoirs in the spleen, are phagocytic immune cells with an ability to assume distinct polarization states in response to environmental stimuli [5]. Polarization of macrophages extends along a continuum of diverse phenotypes with the extreme ends (classically activated pro-inflammatory M1 macrophages and alternatively activated anti-inflammatory M2 macrophages) most commonly recognized [6].

An examination of macrophage cell surface markers in human aneurysmal infrarenal aortic wall sections demonstrated that M1 macrophages are predominant in the aortic adventitia while CD206^+^ (M2) macrophages are localized to the intraluminal thrombus [7]. Studies examining M1/M2 markers in human AAA indicate M1-produced inflammation amplifying cytokines predominate in abdominal aortic aneurysmal disease and suggest a major role for the M1 macrophage phenotype [8]. Macrophages, however, display functional plasticity and possess a remarkable ability to modulate their phenotypic and functional properties in response to tissue-derived cues and signals arising from the local environment [9]. For example, in atherosclerotic disease, the oxidized phospholipid-rich environment directs a macrophage phenotype that is characterized by abundant over-expression of nuclear factor erythroid 2-related factor 2 (Nrf2)-mediated redox-regulatory genes and low phagocytotic capacity [10]. Similarly, in cancer, tumor-associated macrophages are reported to adopt unique phenotypes that facilitate the growth of tumors and aid their subsequent survival [11]. It is thus conceivable that, in AAA, exposure of macrophages to an inflamed, oxidized microenvironment favors the generation of cells with dramatically altered physiology and behavior. While the phenotypes of macrophages in AAA are well documented, the environmentally conditioned responses of these cells to inflammatory stimuli remain to be established. Using a series of case–control experiments, we explored the possibility that the unique AAA microenvironment shapes or directs the responses of differentiated macrophages to persistent inflammatory insults.

## 2. Materials and Methods

All data and materials have been made available at Figshare Knowledge at Figshare.com and can be accessed at: 10.6084/m9.figshare.6291785. The manuscript complies with the Strengthening the Reporting of Observational studies in Epidemiology (STROBE) and Meta-analysis of Observational Studies in Epidemiology (MOOSE) reporting guidelines for observational studies.

### 2.1. Participants and Eligibility Criteria

A case-control study design was used to compare patients with AAA and healthy control participants (Table 1) who were recruited from the Sunshine Coast community between June 2016 and May 2017. Participants included members of the University of the Third Age (U3A) and patients of a Sunshine Coast vascular clinic. Male participants only were recruited to this study as AAA is more prevalent in men than in women and sex differences exist with respect to AAA morphology and hemodynamics. The study was approved by the University of the Sunshine Coast (A/15/705) and the Prince Charles Hospital Human Research Ethics Committees (HREC/16/QPCH/114). All experiments were performed in accordance with relevant guidelines and regulations. Written informed consent was obtained for each participant and all consented participants completed the study. The patient group for biomarker experiments included 19 men with AAA (30–45 mm diameter) and the control group included 36 men without a documented AAA. The patient group for lipid raft and Toll-like receptor 4 (TLR4) imaging experiments included 14 men with AAA (30–45 mm diameter) and the control group included 8 men without a documented AAA. AAA patient aneurysm size was confirmed with ultrasound at study entry. Exclusion criteria for the biomarker experiments were: age < 60 or > 86 years, body mass index (BMI) above 39 kg m^−2^, uncontrolled hypertension (systolic blood pressure (SBP) ≥ 140 mm Hg and diastolic blood pressure (DBP) ≥ 90 mm Hg), cardiac arrhythmia, heart failure, symptomatic aortic stenosis, chronic obstructive pulmonary disease, chronic inflammatory disease and regular use of prescription anti-inflammatory medication. A family history of AAA or known aneurysmal disease served as additional exclusion criteria for control participants. All participants refrained from non-prescribed anti-inflammatory medications 72 h prior to blood collection and abstained from alcohol and caffeine for the 12 h leading up to their visit. 

### 2.2. Monocyte Isolation Protocol

Monocytes were isolated as previously described, using Ficoll-Paque Premium 1.073 (GE Healthcare, Uppsala, Sweden) [12] and hyperosmotic Percoll Plus (Sigma-Aldrich, St. Louis, MO, USA; prepared as described by Repnick et al. [13]) or by discontinuous density centrifugation using isotonic Percoll Plus (Sigma-Aldrich, St. Louis, MO, USA) at densities 1.070 g mL^−1^, 1.062 g mL^−1^, 1.060 g mL^−1^ and 1.058 g mL^−1^. Monocytes within a band located at the interface of the Percoll solution and DPBS/media were collected and diluted with 4 mL of Ca^2+^/Mg^2+^ free DPBS containing 1 mmol L^−1^ EDTA. An aliquot of the cells was stained with May-Grunwald stain (Merck Millipore, Burlington, MA, USA) to confirm the presence of monocytes [14].

### 2.3. Macrophage Maturation and Activation

Monocytes matured into a morphologically heterogeneous macrophage population over 7 days. Macrophages were characterized by excellent purity (98.6 ± 0.6%) and intact activation and phagocytic activities [12]. On day 7, Iscove’s Modified Dulbecco’s Medium (IMDM) and all additives except Macrophage Colony Stimulating Factor (M-CSF)) was replaced for a final time and the adherent macrophages were exposed to 20 ng mL^−1^ interferon gamma (IFN-γ, Sigma-Aldrich, St. Louis, MO, USA) and 0.1 µg mL^−1^
*Escherichia coli*-derived lipopolysaccharide (LPS) (Serotype 0111:B4; Sigma-Aldrich, St. Louis, MO, USA) for 24 h. All supernatants were collected, centrifuged (10,000× *g*, 5 min, 4 °C) and aliquoted with and without butylhydroxytoluene (BHT; 2.5 mg mL^−1^, Sigma-Aldrich, St. Louis, MO, USA). For Western blot analysis, cells were collected in a 300 μL aliquot of radioimmunoprecipitation assay (RIPA) buffer (150 mmol L^−1^ NaCl (BioLab, Clayton, Victoria, Australia), 20 mmol L^−1^ Tris-HCl (Sigma-Aldrich, St. Louis, MO, USA), 1% Igepal CA-630 (Sigma-Aldrich, St. Louis, MO, USA), 0.1% sodium dodecyl sulphate (Sigma-Aldrich, St. Louis, MO, USA), 1% sodium deoxycholate (Sigma-Aldrich, St. Louis, MO, USA), 1 complete protease inhibitor tablet per 7 mL (Roche Diagnostics GmbH, Mannheim, Germany), pH 7.4) and stored at −80 °C until analysis. For glutathione peroxidase (GPx) activity assays, a 300 μL aliquot of GPx collection buffer (50 mmol L^−1^ Tris-HCl, pH 7.5, 5 mmol L^−1^ EDTA (BioLab, Clayton, Victoria, Australia), 1 mmol L^−1^ DTT (Sigma-Aldrich, St. Louis, MO, USA)) was added to each well and adherent cells were harvested and stored as before. For catalase activity assays, a 300 μL aliquot of catalase collection buffer (50 mmol L^−1^ potassium phosphate (BioLab, Clayton, Victoria, Australia), pH 7.0, containing 1 mmol L^−1^ EDTA) was added to each well and adherent cells were harvested and stored as before.

### 2.4. Measurement of Free 8-Isoprostane and a Prostaglandin E_2_ Metabolite

To establish oxidative stress status, free 8-isoprostane levels were measured in BHT-preserved plasma samples and monocyte-derived macrophage supernatants, as described previously [14].

### 2.5. Cytokine and MMP-9 Assays

Cytokines (interleukin-6 (IL-6), transforming growth factor-β (TGF-β), tumor necrosis factor alpha (TNF-α), IL-1β and IL-10) were measured in plasma and monocyte-derived macrophage supernatants and matrix metalloproteinase-9 (MMP-9) levels were measured in plasma using commercially available enzyme immunoassay kits (Affymetrix eBioscience, San Diego, CA, USA) in accordance with manufacturer’s instructions.

### 2.6. Antioxidant Assays

Catalase and GPx enzymatic activities were measured in plasma and monocyte-derived macrophage lysates using commercially available kits (Cayman Chemical Company, Ann Arbor, MI, USA) in accordance with manufacturer’s instructions. Enzyme activities were expressed per µg protein.

### 2.7. Western Blot Analysis

Western blot analysis of monocyte-derived macrophage lysates was carried out to determine expression levels of antioxidant enzymes, MMP-9, TLR4, interleukin-1 receptor-associated kinase 1 (IRAK1) and nuclear factor kappa-light-chain-enhancer of activated B cells (NF-κB). Cell lysates were mechanically homogenized for 2 s at maximum speed (TissueRuptor, QIAGEN PTY LTD Australia, Chadstone, Victoria, Australia) and centrifuged (10,000× *g*, 10 min, 4 °C). The supernatants were collected and balanced for protein concentration. Samples were heated in 2× loading buffer (95 °C; 5 min) and transferred to 7.5% or 12% precast polyacrylamide gels (Bio-Rad Laboratories, Gladesville, NSW, Australia). Proteins were separated by electrophoresis (150 V, 48–70 min, 22 °C) in running buffer (25 mmol L^−1^ Tris base (Sigma-Aldrich, St. Louis, MO, USA), 192 mmol L^−1^ glycine (Sigma-Aldrich, St. Louis, MO, USA), 0.1% SDS (Sigma-Aldrich, St. Louis, MO, USA), pH ~8.3 without adjustment) and transferred to Immobilon-P polyvinylidene difluoride (PVDF) membranes using a semi-dry transfer cell (Bio-Rad Laboratories, Gladesville, NSW, Australia) at 15 V for 15 min. The membranes were blocked with 3% bovine serum albumin (BSA; Sigma-Aldrich, St. Louis, MO, USA) or 5% non-fat milk in Tris-buffered saline (pH 7.6) with 0.1% Tween-20 for 60 min followed by overnight incubation with antibodies to catalase (Abcam, Cambridge, UK, rabbit polyclonal, IgG, titer 2 µg mL^−1^, catalog number ab16731), GPx (Cell Signaling Technology, Danvers, MA, USA, rabbit polyclonal, titer 1:50 dilution of stock, catalog number 3206), MMP-9 (Cell Signaling Technology, Danvers, MA, USA, rabbit polyclonal, titer 8.1 µg mL^−1^, catalog number 3852), TLR4 (Santa Cruz Biotechnology, Santa Cruz, CA, USA, mouse monoclonal, IgG_1_ (kappa light chain), titer 0.5 µg mL^−1^, catalog number SC-293072), NF-κB (Cell Signaling Technology, rabbit IgG, titer 1:400 dilution of stock, catalog number 3033), IRAK1 (Cell Signaling Technology, rabbit IgG, titer 1:400 dilution of stock, catalog number 4504) or glyceraldehyde 3-phosphate dehydrogenase (GAPDH; loading control; Abcam, mouse monoclonal [6C5], IgG_1_, titer 2.5 µg mL^−1^, catalog number ab8245). Bands were visualized using an IgG Vectastain kit (Vector Laboratories Inc., Burlingame, CA, USA) and 3,3′-diaminobenzidine tetrahydrochloride (DAB) with metal enhancer (Sigma-Aldrich, St. Louis, MO, USA). Image J software (NIH, Bethesda, MD, USA) was used to quantify band intensity, with subtraction of background intensity from an adjacent region of the membrane.

### 2.8. Confocal Microscopy

Human monocytes (1.3 × 10^5^ cells per well) were seeded onto 8-well chamber slides (Sarstedt Australia, Mawson Lakes, SA, Australia) and maintained in culture as described above. On day 7, adherent macrophages were primed with 0.1 µg mL^−1^ LPS and 20 ng mL^−1^ IFN-γ for 24 h. At the end of the priming period, culture supernatants were removed, and the cells were exposed to LPS and IFN-γ for a second time (4 h). Cells were washed once with serum-free medium and labelled with cholera toxin subunit B conjugated to Alexa Fluor 488 (Thermo Fisher Scientific, Seventeen Mile Rocks, Queensland, Australia, catalog number C34775; 4 µg mL^−1^) for 20 min at 4 °C. Following three rinses with serum-free medium, the cholera toxin subunit B was cross-linked with cholera toxin B-subunit antibody (1:200 dilution, Millipore, goat polyclonal, IgG, Catalog number 227040; 15 min, 4 °C). Cells were rinsed, fixed using formalin (3.7%, 5 min, 4 °C) and methanol (100%, 5 min, −20 °C), washed with PBS (3 × 5 min, 4 °C) and incubated with Image iT Fx signal enhancer (Life Technologies, Mulgrave, Victoria, Australia; 30 min, room temperature). Cells were rinsed three times with PBS and incubated overnight (4 °C) with anti-TLR4 mAb (Santa Cruz Biotechnology, mouse monoclonal, IgG_1_ (kappa light chain), titer 0.2 µg mL^−1^, catalog number SC-293072; 0.2 µg mL^−1^) and DAPI (Sigma-Aldrich, St. Louis, MO, USA, catalog number D9542-IMG; 2 µg mL^−1^). On the following day, cells were washed with PBS (3 × 5 min, 4 °C) and incubated with goat anti-mouse IgG secondary antibody, Alexa Fluor 568 (Thermo Fisher Scientific, F(ab′) 2-Goat anti-mouse IgG (H + L) cross-adsorbed secondary antibody, titer 10 µg mL^−1^, catalog number A11019; 1 h, 4 °C). Cells were mounted with glycerol and immobilized beneath glass coverslips. Confocal images were acquired using a Nikon digital sight DS-QilMc monochrome camera (square pixel CCD, 1.5 megapixels; Nikon Corporation, Tokyo, Japan) attached to a Nikon Eclipse T*i* microscope. Redistribution of the pro-inflammatory LPS receptor, TLR4, was investigated and cells used in these analyses were selected based on DAPI staining without reference to TLR4 immunofluorescence to avoid bias. TLR4 antibody was validated using Western blot analysis. Only a single band was detected at the appropriate molecular weight (95 kDa). No fluorescence was detected by confocal microscopy when the primary antibody was omitted.

### 2.9. RNA Isolation and Real-Time Quantitative PCR

RNA isolation and real-time quantitative PCR was carried out as described by Meital et al. [12]. Primers for TLR4, Nrf2 and glyceraldehyde 3-phosphate dehydrogenase (GAPDH) were as described by Song et al. [15], Saw et al. [16], and Long et al. [17] respectively. All samples were analyzed in duplicate and standards were analyzed in triplicate (technical replicate).

### 2.10. Data Analysis

Group size estimates of 6 (8-isoprostane) and 31 (PGE_2_) were calculated with 80% power (alpha level of 0.05) using power/sample size (Univ. British Columbia, Vancouver, BC, Canada) and pooled variance (Solvers statistics) calculators. Values were based on previously reported levels in healthy controls (8-isoprostane, 32 ± 15.0 pg mL^−1^; PGE_2_, 19.9 ± 14.5 pg mL^−1^) and patients with AAA (8-isoprostane, 56.0 ± 8.9 pg mL^−1^; PGE_2_, 38.8 ± 34.35 pg mL^−1^) [18,19]. Continuous demographic data for AAA patients and control participants were compared using a Student’s *t*-test and are presented as mean ± SD. Categorical demographic variables were compared using a Fisher’s exact test. Data were assessed to determine normality (Shapiro–Wilk test) and homogeneity of variance (Levene statistic). Experimental data are presented as median and between group differences were examined by Mann-Whitney U analysis. Outliers in plasma cytokine data (TNF-α *n* = 2, IL-6 *n* = 1 and TGF-β *n* = 1) were identified using the extreme Studentized deviate many-outlier procedure [20] and were excluded from analysis. Mantel-Haenszel analysis revealed no confounding influence of comorbidities (hypertension, hyperlipidemia and coronary heart disease) or drug therapies (β-adrenoceptor antagonists, statins and low dose aspirin) for macrophage supernatant TNF-α and IL-6 data (Table 2). Data were analyzed with Prism (GraphPad Software, La Jolla, CA, USA) and statistical significance was set at *p* < 0.05.

## 3. Results

### 3.1. AAA Was Associated with Systemic Inflammation

Plasma concentrations of 8-isoprostane (Figure 1a) and TNF-α (Figure 1b) were markedly elevated in AAA patients compared to control participants (*p* = 0.0018 and 0.0170). In contrast, concentrations of IL-6 (Figure 1c), TGF-β (Figure 1d) and MMP-9 (Figure 1e) in AAA plasma were comparable to the control cohort. The plasma concentration of a PGE_2_ metabolite was markedly lower in patients with AAA compared to control participants (Figure 1f; *p* = 0.014), as was activity of the antioxidant enzyme catalase (Figure 1g; *p* = 0.0096). GPx activity (Figure 1h) was similar across groups.

### 3.2. Macrophage Responses to LPS Stimulation Were Suppressed in AAA

LPS-stimulated 8-isoprostane production, expressed as a function of total number of circulating monocytes present in whole blood, was markedly suppressed in AAA macrophage supernatants (Figure 2b) compared to control participants (Figure 2a) with the slope of the control data (*R*^2^ = 0.93) 9.1-fold greater than that of the AAA data (*R*^2^ = 0.13, *p* = 0.0063). In addition, AAA macrophages produced lower levels of TNF-α (Figure 2c) and IL-6 (Figure 2d) in response to LPS stimulation compared to control participants (*p* = 0.0214 and 0.0045, respectively). Protein levels of both proMMP-9 (Figure 2f) and mature MMP-9 (Figure 2g) were markedly reduced in macrophage lysates from AAA patients (35% and 40% of control band intensity, respectively; *p* = 0.0095 and 0.0381). A non-significant trend for higher levels of TGF-β was observed in supernatants of macrophages from AAA patients (Figure 2e) compared to control participants. Concentrations of IL-10 and IL-1β in AAA macrophage supernatants were not different to control (IL-10, 0.22 ± 0.09 vs. 0.26 ± 0.05 ng mL^−1^; IL−1β, 3.3 ± 1.14 vs. 2.0 ± 0.46 pg mL^−1^). Catalase (Figure 2h) and GPx activity (Figure 2j) in LPS-stimulated AAA macrophage lysates was similar to control participants and no difference in the protein expression of either enzyme (Figure 2i,k) was observed for the two cohorts.

### 3.3. Nrf2 mRNA Transcripts Were Increased in AAA Macrophages

Nuclear factor erythroid 2-related factor 2 mRNA levels were significantly higher in both non-stimulated (*p* = 0.027) and 24 h LPS-stimulated macrophages (*p* = 0.003) from patients with AAA compared to control (Figure 3).

### 3.4. Lipid Raft and TLR4 Internalization Was Observed in AAA Macrophages

In control participants, in the absence of LPS stimulation (Figure 4a,b) and following 40 min LPS stimulation (Figure 4e,g), lipid rafts were localized to the plasma membrane of macrophages with no evidence of lipid raft internalization. Following a 28 h exposure to LPS (Figure 4i,j), lipid raft internalization was observed in all cells. In AAA macrophages, lipid raft internalization was observed in both the non-stimulated (Figure 4m,n) and LPS-stimulated (Figure 4q,r) conditions. TLR4 was localized to the plasma membrane of non-stimulated (Figure 4a,c) and 40 min LPS-stimulated (Figure 4e,g) control macrophages but not AAA macrophages in either the non-stimulated (Figure 4m,o) or LPS-stimulated (Figure 4q,s) conditions. TLR4 was diffusely localized within the cytosol in 28 h LPS-stimulated macrophages from control (Figure 4i,k) and AAA (Figure 4q,s) participants. AAA macrophage lysate TLR4 mRNA (Figure 4u) and protein concentration (Figure 4v) were similar to the control.

### 3.5. Levels of IRAK1 and Phosphorylated NF-κB Were Significantly Decreased in AAA

Protein levels of interleukin-1 receptor-associated kinase 1 (IRAK1) were lower in non-stimulated macrophages from patients with AAA (*n* = 9) compared to control (*n* = 6; *p* = 0.02) (Figure 5a). IRAK1 mRNA transcript levels were not different between the groups (Figure 5b). Levels of phosphorylated NF-κB p65 protein subunit were significantly lower in non-stimulated macrophages from patients with AAA (*n* = 11) compared to control (*n* = 7; *p* = 0.04) (Figure 5c).

## 4. Discussion

Systemic inflammation and oxidative stress are hallmark features of AAA and are well-documented to underlie the various pathological processes contributing to the progression of this disease [21]. In support of this, our data showed increased circulating levels of TNF-α and the oxidative stress biomarker 8-isoprostane and reduced activity of the antioxidant enzyme catalase in plasma samples from patients with AAA. Elevated levels of TNF-α have consistently been identified in AAA tissue [22], plasma [23] and serum [24] and there is evidence that the concentration of this cytokine is higher in patients with small compared to large AAA [25]. In contrast, the association between 8-isoprostane levels and AAA or catalase activity and AAA has been examined in few studies involving human participants despite compelling evidence linking impaired systemic redox status to multiple AAA mechanical and molecular events [26,27]. A study by Pincemail et al. [28] identified an 18% increase in urinary levels of 8-isoprostane in patients with small AAA and a 66% increase in patients with large AAA (>50 mm) compared to an age-matched non-AAA control group. Similarly, Ramos-Mozo and colleagues [29] reported markedly decreased catalase activity levels in plasma samples obtained from 103 asymptomatic patients with small (*n* = 56) and large (*n* = 47) AAA compared to a non-AAA control cohort (*n* = 34). Taken together, the results obtained add to accruing evidence that AAA is a chronic vascular disease underpinned by systemic inflammation and a pathological state of oxidative stress.

Environmental cues shape adaptive macrophage responses that either induce resolution and repair mechanisms or allow progression toward a protracted disease state. Endotoxin tolerance, which is a prime example of such shaping, mitigates against tissue damage caused by excessive activation of macrophages in an acute setting [30]. In AAA, it is conceivable that exposure to LPS-secreting Gram negative bacteria [31], cleaved extracellular matrix components [32] or an increased circulating concentration of fibrinogen [33], provide a level of endogenous danger signaling sufficient to prime macrophage TLR4 towards an endotoxin tolerant state. In line with this hypothesis, our study provides the first evidence that AAA macrophages exhibit a refractory phenotype characterized by suppressed production of inflammatory cytokines, oxidative stress biomarkers and elastolytic enzymes following LPS challenge. The abrogated TNF-α and IL-6 response and downregulation of MMP-9 identified in AAA macrophages is characteristic of endotoxin tolerance as described by evidence from human studies [34,35]. Data arising from microarray studies [36,37] indicate that a subset of LPS response genes, termed non-tolerizeable genes (e.g., IL-1β), show equivalent or upregulated expression levels following re-exposure to inflammatory stimuli. Consistent with this, AAA monocyte-derived macrophages produced similar levels of IL-10 and IL-1β compared to a control cohort and showed a trend toward increased production of TGF-β. The combined evidence suggests that the inflamed, oxidized AAA microenvironment skews macrophages toward an immunosuppressed endotoxin tolerant-like phenotype that is less equipped to stimulate resolution-directed innate immune responses.

While endotoxin tolerance has been reported to protect against uncontrolled systemic inflammation, tissue damage and septic shock lethality in animal models of disease [38,39], there is evidence that macrophage tolerization may be deleterious in the context of human disease. For example, chronic suppression of the immune response in pathologies such as sepsis can lead to long-lasting inflammation, opportunistic infection and, in some instances, death [40,41]. In sepsis patients in particular, decreased monocyte human leukocyte antigen-DR (HLA-DR) expression, an established surrogate marker of immunosuppression, has consistently been shown to independently predict the occurrence of nosocomial infection and mortality [42,43]. While the impact of endotoxin tolerant-like immunosuppression on AAA disease progression has yet to be determined, a growing number of reports have linked diminished immune capacity in AAA with adverse clinical disease outcomes. For example, preventive IL-6 neutralization in an animal model of AAA increased 7-day mortality by >40% [44] and studies examining the disease course of AAA in organ transplant patients have reported aggressive expansion and accelerated rupture rates following commencement of immunosuppressive treatment regimen [45,46].

This study examined tolerizing mechanisms in monocyte-derived macrophages at the mRNA, receptor and lipid raft level. We measured Nrf2 mRNA levels in macrophages from AAA patients and control participants in light of recent evidence suggesting this transcription factor negatively regulates macrophage responses to LPS [47]. We found significantly upregulated Nrf2 mRNA levels in both non-stimulated and LPS-stimulated macrophages from patients with AAA compared to the control. As Nrf2 is reported to inhibit pro-inflammatory cytokine gene expression in M1 macrophages through direct binding to DNA, it is possible that this transcription factor represents a tolerizing mechanism in AAA. We next investigated the distribution of TLR4 and lipid rafts in macrophages isolated from patients with AAA using confocal microscopy. We found that TLR4 and lipid raft localization was diffusely cytosolic in AAA macrophages in both the non-stimulated and LPS-stimulated conditions whereas in control participant macrophages, these were mainly localized to the plasma membrane. It is well established that ligand-induced activation of TLR4 occurs within cholesterol- and sphingolipid-enriched membrane rafts in concert with the co-receptor, cluster of differentiation 14 (CD14), at the plasma membrane [48]. This makes it likely that the low association of TLR4 and lipid rafts with the plasma membrane in AAA macrophages contributes to the suppressed response of these innate immune cells to stimulation by LPS. The abrogated TLR4 signaling that accompanies tolerance in cells of human origin has previously been shown to occur in the absence of downregulation of the receptor [49]. In line with this, our study found similar TLR4 mRNA and protein levels in AAA macrophages compared to the control.

To further explore the phenomenon of attenuated signaling within this pro-inflammatory pathway, mRNA and protein levels of the kinase IRAK1 and protein levels of NF-κB, a transcription factor downstream of TLR4 were examined. IRAK1 is activated by the TLR4 adaptor MyD88 and diminished levels of this protein have a crucial role in endotoxin tolerance [50]. In addition, decreased phosphorylation of NF-κB p65 subunit is strongly associated with endotoxin tolerance [51,52]. Lower IRAK1 protein levels and reduced NF-κB phosphorylation observed in this study concur with the observed abrogated TLR4 signaling in AAA and highlight these as potential mechanisms involved in tolerance induction.

The lower levels of PGE_2_ in AAA patient plasma are of interest. PGE_2_ mediates initiation of inflammatory responses, moderation of inflammatory response magnitude and duration and, in later stages, activation of eicosanoid switching toward pathways of resolution and healing [53]. Decreased PGE_2_ production in AAA raises the possibility that impaired eicosanoid switching may be a contributory factor in the non-resolving inflammation that is a hallmark feature of AAA.

### Limitations

A limitation of this study was the absence of ultrasound screening for AAA in control participants. While an in vitro cell culture system is subject to inherent limitations, our use of native, non-immortalized and non-transformed primary cells allowed a biologically relevant representation of the unique, cell-specific responses of AAA macrophages to an LPS stimulus. In addition, the functional changes/responses observed for monocyte-derived macrophages are likely to be indicative of macrophages within the AAA aorta as aneurysmal macrophages derive primarily from a circulating monocyte precursor. The evidence for endotoxin tolerance in patients with AAA was obtained by examining blood-derived macrophages in a cell culture environment and, while phenotypic changes are possible during cell culture, comparisons between patients with AAA and control subjects were made using cells that were grown under identical conditions. In addition, a previous study examining macrophage subtypes has reported that cells isolated by laser capture microdissection in AAA tissue demonstrate similar gene expression profiles to those obtained in vitro [7], suggesting that cellular changes occurring during or after cell harvest may be minor. While MMP-9 protein concentration and Nrf2 mRNA expression was reported for macrophage lysates, further studies could correlate these end-points with their activity. Further studies are also warranted to examine evidence of endotoxin tolerance in human AAA macrophages in situ, and to investigate whether similar responses are observed in macrophages obtained from female patients who have AAA.

## 5. Conclusions

Taken together, the results reported here indicate that, against a background of chronic systemic inflammation and oxidative stress, AAA macrophages adopt an endotoxin tolerant-like phenotype that is characterized by a diminished capacity to produce mediators that enhance the immune response. Diminished immune capacity in AAA may exacerbate the persistent inflammation and consequent failure of resolution that underlies this disease state. In this setting, strategies aimed at reversing tolerance may provide some benefit. Tolerance thus represents an area of investigation with the potential to yield biologically plausible therapeutic targets.

## Figures and Tables

**Figure 1 antioxidants-09-00896-f001:**
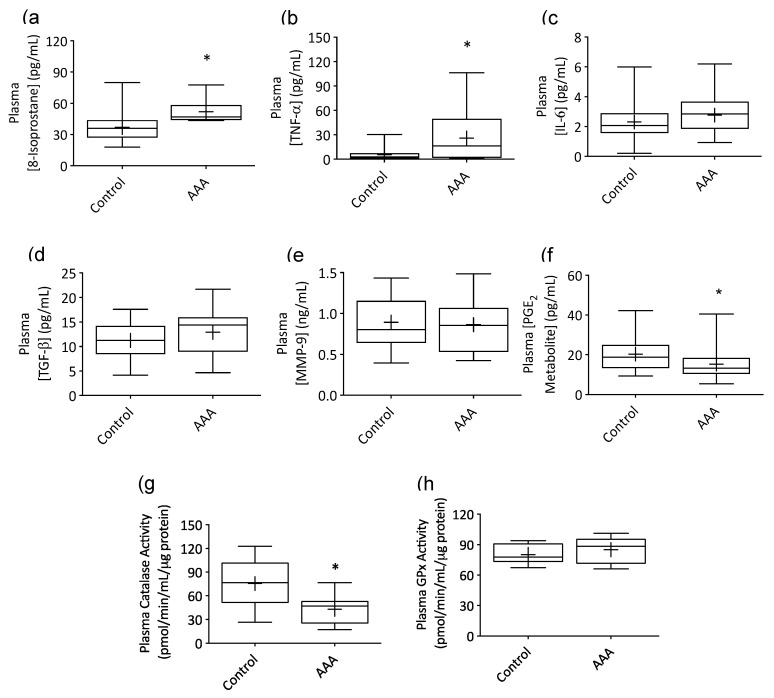
Plasma 8-isoprostane and cytokine concentrations and antioxidant enzyme activity in patients with small AAA and healthy control participants. Plasma 8-Isoprostane (**a**); AAA *n* = 10, control *n* = 17) and tumor necrosis factor alpha (TNF-α; (**b**); AAA *n* = 19, control *n* = 34) concentrations were markedly elevated and catalase activity (**g**); AAA *n* = 9, control *n* = 16) and PGE_2_ metabolite levels (**f**); AAA *n* = 30, control *n* = 34) significantly reduced in AAA patients compared to control participants. Interleukin-6 (IL-6; (**c**), transforming growth factor-β (TGF-β; (**d**), matrix metalloproteinase-9 (MMP-9; (**e**) concentrations and glutathione peroxidase activity (GPx; (**h**) were similar across groups. Box plots show 25th, 50th (median) and 75th percentiles (horizontal lines), mean (+) and minimum and maximum values (whiskers). * *p* < 0.05 (Mann-Whitney analysis).

**Figure 2 antioxidants-09-00896-f002:**
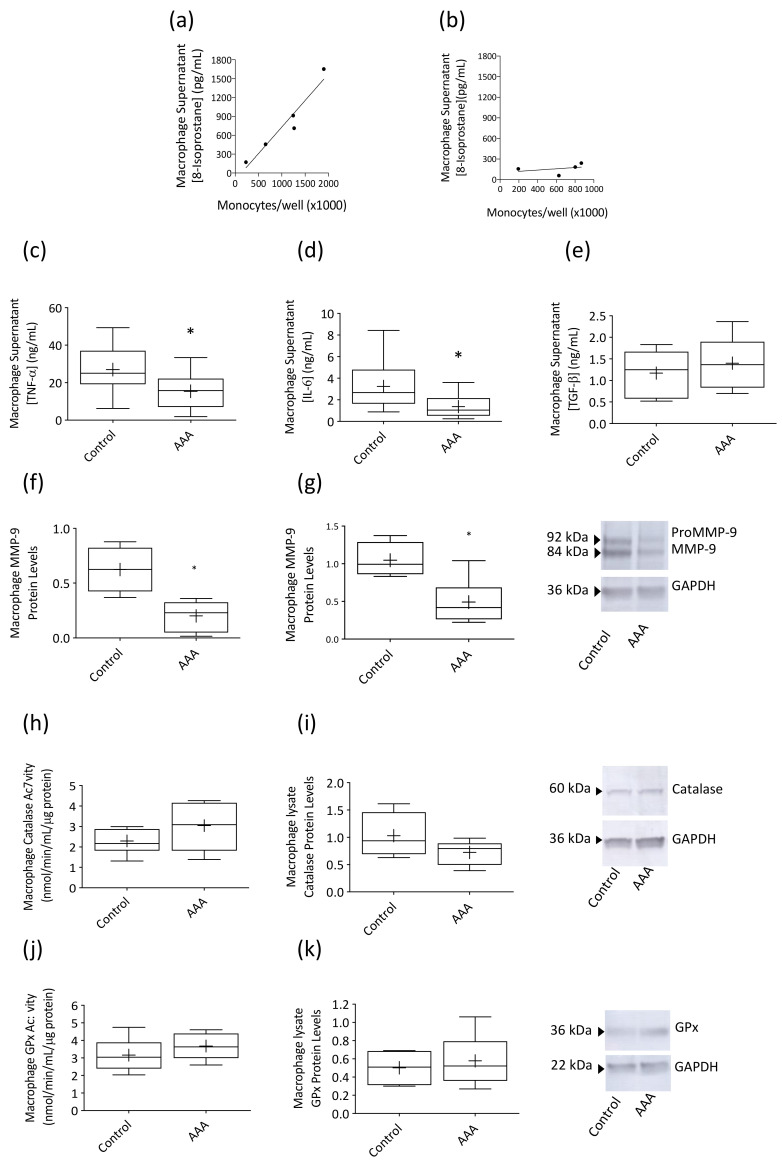
Cytokine, 8-isoprostane and antioxidant enzyme activity levels in supernatants and lysates obtained from AAA patient and control subject monocyte-derived macrophages. LPS-stimulated 8-isoprostane production, expressed as a function of total number of circulating monocytes present in whole blood samples, was markedly suppressed in AAA macrophage supernatants (**b**); *n* = 4) compared to control participants (**a**); *n* = 5). AAA macrophages also produced lower concentrations of tumor necrosis factor alpha (TNF-α; (**c**); *n* = 9) and interleukin-6 (IL-6; (**d**); *n* = 12) compared to control macrophages (*n* = 17 and *n* = 14) and both pro-matrix metalloproteinase-9 (ProMMP-9; 92 kDa, (**f**)) and mature MMP-9 (84 kDa, (**g**)) protein levels were significantly reduced (both *n* = 6). Transforming growth factor-β (TGF-β; (**e**)) concentrations tended to be higher in AAA macrophage supernatants (AAA, *n* = 10; control, *n* = 14), although this did not achieve significance. Catalase activity ((**h**); AAA, *n* = 7; control, *n* = 7) and expression ((**i**); AAA, *n* = 6; control, *n* = 4) and glutathione peroxidase (GPx) activity ((**j**); AAA, *n* = 8; control, *n* = 8) and expression ((**k**); AAA, *n* = 6; control, *n* = 4) were similar to the control cohort. Box plots show 25th, 50th (median) and 75th percentiles (horizontal lines), mean (+) and minimum and maximum values (whiskers). * *p* < 0.05 (Mann–Whitney analysis).

**Figure 3 antioxidants-09-00896-f003:**
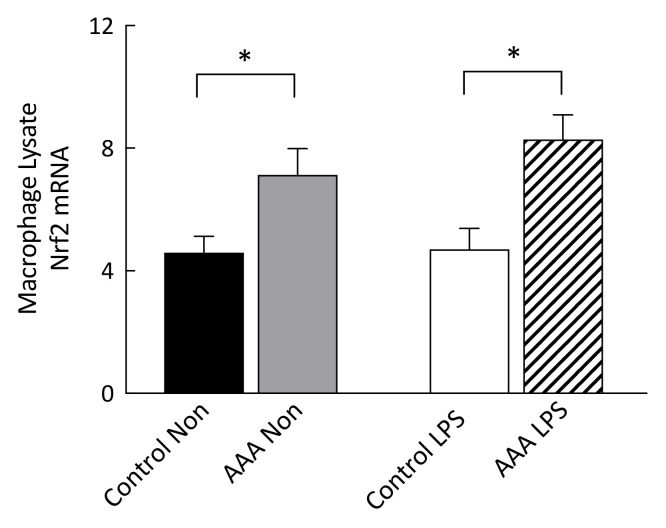
Comparison of relative levels of Nrf2 transcripts in macrophages from control participants and patients with AAA. Nrf2 mRNA levels, measured by real time qPCR, were normalized to GAPDH. Nrf2 mRNA was significantly increased in both non-stimulated and 24 h LPS-stimulated (*n* = 8) AAA macrophages compared to control (*n* = 9). * *p* < 0.05.

**Figure 4 antioxidants-09-00896-f004:**
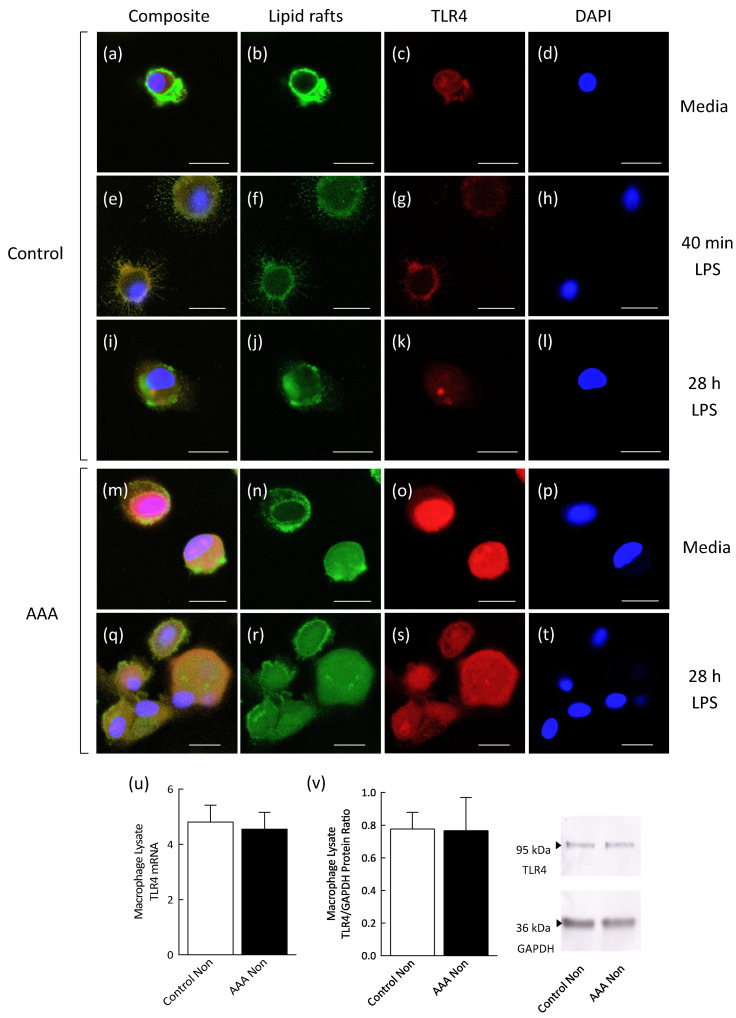
Representative images showing differential distribution of lipid rafts and TLR4 in human monocyte-derived macrophages before (control, **a**–**d**; AAA, **m**–**p**) and after 40 min LPS stimulation (control, **e**–**h**) or 28 h LPS stimulation (control, **i**–**l**; AAA, **q**–**t**). Lipid rafts were localized to the membrane of non-stimulated (**a**,**b**) and 40 min LPS-stimulated macrophages from *n* = 8/8 control participants (**e**,**f**) and lipid raft internalization was observed in *n* = 8/8 control participant macrophages following 28 h LPS stimulation (**i**,**j**). Lipid rafts were internalized in macrophages from *n* = 7/10 AAA participants in the non-stimulated condition (**m**,**n**) and in *n* = 10/10 AAA participant macrophages following exposure to 28 h LPS (**q**,**r**). TLR4 was localized to the membrane of non-stimulated (**a**,**c**) and 40 min LPS-stimulated control macrophages (**e**,**g**). TLR4 was associated with the cytosol and membrane of control macrophages exposed to LPS for 28 h (**i**,**k**). TLR4 was primarily internalized in non-stimulated (**m**,**o**) and 28 h LPS-stimulated macrophages from the AAA participants (**q**,**s**). AAA macrophage lysate TLR4 mRNA (**u**; AAA, *n* = 9; control, *n* = 8) and protein levels (**v**; AAA, *n* = 7; control, *n* = 5) were similar to the control. TLR4 mRNA and protein levels were normalized to GAPDH. No fluorescence was detected by confocal microscopy when the primary antibody was omitted (data not shown). Red—Alexa Fluor 568-labelled TLR4; blue—DAPI; green—fluorescein isothiocyanate (FITC)-labelled cholera toxin subunit B (detects GM1 ganglioside lipid raft marker). Scale bar = 20 µm.

**Figure 5 antioxidants-09-00896-f005:**
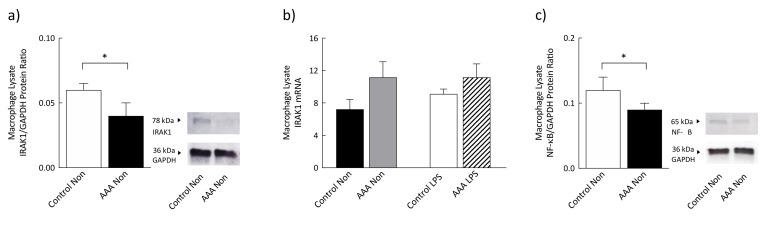
Protein (**a**) and mRNA transcript levels of interleukin-1 receptor-associated kinase 1 (IRAK1) (**b**) and phosphorylation levels of the p65 protein subunit of NF-κB (**c**) in control and AAA monocyte-derived macrophages. Values are mean ± SEM. * *p* < 0.05.

**Table 1 antioxidants-09-00896-t001:** Demographic, biometric and medical characteristics of abdominal aortic aneurysm (AAA) patients and control participants. Continuous demographic data are presented as mean ± SD, categorical demographic data are presented as number (percentage).

	Biomarker Experiments		Confocal Experiments	
Variable	AAA Patients (*n* = 19)	Control Participants (*n* = 36)	AAA Patients (*n* = 14)	Control Participants (*n* = 8)
Gender (Male/Female)	19/0	36/0	14/0	8/0
Age (years)	74.6 ± 5.8	71.7 ± 5.2	73.0 ± 4.92	69.7 ± 5.15
AAA size (mm)	37.8 ± 5.3		37.0 ± 5.5	
Smoking: Never	7	16	3	3
Past	10	20	9	4
Current	2	0	2	1
BMI (kg/m^2^)	26.9 ± 2.9	25.8 ± 3.4	27.5 ± 3.6	24.9 ± 4.0
Systolic blood pressure (mmHg)	136.5 ± 13.6	135.9 ± 11.3	142.1 ± 14.7	131.3 ± 14.4
Diastolic blood pressure (mmHg)	78.6 ± 6.1	79.6 ± 8.3	81.0 ± 7.6	84.0 ± 4.2
Coronary heart disease	12 (63%) †	3 (8%)	5 (35%)	0 (0%)
Diabetes	2 (11%)	0 (0%)	0 (0%)	0 (0%)
Dyslipidemia	15(79%) †	9 (25%)	9 (64%)	2 (25%)
Hypertension	12 (63%)	13 (36%)	9 (64%)	3 (38%)
NSAIDs	5 (26%)	3 (8%)	2 (14%)	0 (0%)
Ace inhibitors	2 (11%)	2 (6%)	2 (14%)	1 (13%)
Alpha-1 blockers	0 (0%)	2 (6%)	0 (0%)	0 (0%)
AT receptor antagonists	6 (32%)	6 (17%)	3 (21%)	2 (25%)
Beta blockers	8 (42%) †	4 (11%)	2 (14%)	2 (25%)
Calcium channel blockers	2 (11%)	3 (8%)	3 (21%)	2 (25%)
Diuretics	2 (11%)	1 (3%)	0 (0%)	0 (0%)
Anti-platelet drugs	10 (53%) †	2 (6%)	7 (50%)	1 (13%)
Statins	14 (74%) †	10 (28%)	9 (64%)	2 (25%)

† AAA significantly different to control (Fisher’s exact test, *p* < 0.05). Prostaglandin PGE_2_ data were measured using plasma from both AAA cohorts.

**Table 2 antioxidants-09-00896-t002:** Mantel-Haenszel analysis of potential confounding variables.

Endpoint	Variable	Chi Square	d.f.	*p*	Confounder
TNF-α	Beta blockers	0.19	1	0.66	No
TNF-α	Statins	1.18	1	0.28	No
TNF-α	Low dose Aspirin	0.10	1	0.75	No
TNF-α	Hyperlipidemia	0.56	1	0.46	No
TNF-α	Hypertension	0.01	1	0.91	No
IL-6	Beta blockers	0.23	1	0.63	No
IL-6	Statins	0.00	1	0.96	No
IL-6	Hyperlipidemia	0.12	1	0.73	No
IL-6	Hypertension	0.17	1	0.68	No
IL-6	Coronary heart disease	0.66	1	0.42	No

d.f. Degrees of freedom.

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
