# Peer review of "Endotoxin Tolerance in Abdominal Aortic Aneurysm Macrophages, In Vitro: A Case–Control Study"

_antioxidants, 2020, doi:10.3390/antiox9090896_

Round 1
Reviewer 1 Report
The manuscript by Meital et al. describes the observation of endotoxin tolerance in macrophages isolated from patients with abdominal aortic aneurysm. The topic is very interesting especially in view of a still missing therapy for AAA. The data are from in vitro experiments. The present paper is very well written, the illustrations are clear and the conclusions are based on the generated data. However, the following points must be taken into account:
Minor comments
- few spelling mistakes, e.g. line 285 missing word – please check throughout the whole manuscript.
Major comments
- If I understand correctly, the study involves only male subjects, what is the reason?
- Why were patients under 60 years of age excluded from the study?
- Figure 4, immunofluorescence staining: please show a double staining of lipid drafts and TLR4 expression to show the correlation
- The authors also point out the limitations of the study, perhaps this paragraph should also be marked as such.
- Figure 4: how often were the immunofluorescence stains repeated or how many different patients were analysed?
- The western blot images are of poor quality; would you please insert better ones.
- In your results, do you see any differences in the reactivity of macrophages with respect to different therapy in AAA patients?
- The effect of enzymes (such as MMP9) or transcription factors (such as Nrf2) is based not only on their gene or protein expression, but also on their activity. To be able to make a statement about the role of these factors, you also need the corresponding activity data. Please insert them.
Author Response
Reviewer #1
The manuscript by Meital et al. describes the observation of endotoxin tolerance in macrophages isolated from patients with abdominal aortic aneurysm. The topic is very interesting especially in view of a still missing therapy for AAA. The data are from in vitro experiments. The present paper is very well written, the illustrations are clear and the conclusions are based on the generated data. However, the following points must be taken into account:
Minor comments
- few spelling mistakes, e.g. line 285 missing word – please check throughout the whole manuscript.
Response: On Page 10, Line 260, we added the word “that” between “possible” and “this”. We have checked the manuscript and made all necessary corrections, including use of American spelling throughout.
Major comments
- If I understand correctly, the study involves only male subjects, what is the reason?
Response: The risk of AAA is higher in men than in women (odds ratio, 3.24; Summers et al., 2020). Considerable evidence exists for sex differences in morphology and hemodynamic responses in AAA. Morphologically, women have a smaller volume of intraluminal thrombus than men (Gao et al., 2020). Hemodynamic differences include higher peak pressure and higher wall shear stress at the lower anterior area of the AAA, and lower oscillatory shear stress index for women compared to men (Gao et al., 2020). For patients with AAA, the risk of rupture is significantly greater for women than for men (Brown et al., 1999).
In this study, we isolated monocytes from whole blood and allowed spontaneous differentiation to macrophages in culture. Matrix metalloproteinase MMP-9 and cytokine levels were examined in the cells. mRNA expression of MMP-9 in non-thrombus-covered aneurysm wall has been reported to be significantly higher in women than in men with AAA (Villard et al., 2014). In an elastase-perfusion rat model of AAA, plasma concentrations of cytokines and chemokines (bone morphogenic protein, interleukins 1, 2, 3, 5, 7, 11 and 12; transforming growth factor β1 and β2; and vascular endothelial growth factors 1 and 2), were significantly different between male and female rats (Sinha et al., 2006).
Male only patients were included in this study to avoid a potential confounding influence of sex on measurements of cytokines and MMP-9. Further studies would be required to determine whether tolerance also exists in women with AAA.
We have included the following sentence in the Method section, Page 2, Lines 79-81: “Male participants only were recruited to this study as AAA is more prevalent in men than in women and sex differences exist with respect to AAA morphology and hemodynamics.”
Brown, L.C. and Powell, J.T. Risk factors for aneurysm rupture in patients kept under ultrasound surveillance. UK small aneurysm trial participants. Ann. Surg., 1999;230:289-296.
Gao, Z. et al., Gender differences of morphological and hemodynamic characteristics of abdominal aortic aneurysm. Biology of Sex Differences, 2020;11:41.
Sinha, I. et al., Female gender attenuates cytokine and chemokine expression and leukocyte recruitment in experimental rodent abdominal aortic aneurysms. Annals New York Academy Sciences, 2006;1085:367-379.
Summers, K. et al., Evaluating the prevalence of abdominal aortic aneurysms in the United States through a national screening database. Journal of Vascular Surgery, 2020, S0741-5214.
Villard, C. et al., Differences in elastin and elastolytic enzymes between men and women with abdominal aortic aneurysm. Aorta, 2014;2:179-185.
- Why were patients under 60 years of age excluded from the study?
Response: The concentration of many of the biomarkers tested in this study are known to be affected by age (Álvarez-Rodrίguez et al., 2012; Lee et al., 2017). For example, the plasma concentration of IL-6 was significantly higher in healthy people who were aged >60 years compared to those who were <30 years and those who were 30-60 years (Álvarez-Rodrίguez et al., 2012). Patients under the age of 60 were excluded from our study to minimize the potential of age as a confounding variable. Age of patients with AAA and controls was not significantly different (Table 1).
Álvarez-Rodrίguez, L. et al., Aging is associated with circulating cytokine dysregulation. Cellular Immunology, 2012;273:124-132.
Lee, D.H. et al., Age-dependent alterations in serum cytokines, peripheral blood mononuclear cell cytokine production, natural killer cell activity, and prostaglandin F2α. Immunologic Research, 2017;65:1009-1016.
- Figure 4, immunofluorescence staining: please show a double staining of lipid drafts and TLR4 expression to show the correlation
Response: Thank you for this suggestion. We have now revised Figure 4 (Page 9, Lines 177-189) to include triple-immunofluorescence staining (DAPI, lipid rafts, TLR4) to highlight the relative distributions of these markers in the macrophages.
- The authors also point out the limitations of the study, perhaps this paragraph should also be marked as such.
Response: Page 11, Line 287: We have added a subheading, 4.1 Limitations.
- Figure 4: how often were the immunofluorescence stains repeated or how many different patients were analysed?
Response: Immunofluorescence staining was carried out once per macrophage sample for each participant, with n=8 control participants and n=10 AAA patients. N value were included in the both Results text and in the Figure 4 legend.
- The western blot images are of poor quality; would you please insert better ones.
Response: Western blot images reflect quite low protein expression levels, as evidenced by differences in band intensity compared with the housekeeper, GAPDH. Nonetheless, bands were readily detectable and could be easily quantitated using ImageJ software.
- In your results, do you see any differences in the reactivity of macrophages with respect to different therapy in AAA patients?
Response: Macrophage pro-inflammatory cytokine production (TNFα and IL-6) in response to a proinflammatory stimulus (exposure to lipopolysaccharide) is indicative of macrophage reactivity. We analysed the potential confounding influence of co-morbidities (hypertension, hyperlipidemia, coronary heart disease) or drug therapies (β-adrenoceptor antagonists, statins, and low-dose aspirin), which were shown to be significantly different between the AAA and control cohorts (Table 1). A Mantel-Haenszel analysis detected no confounding influence for these parameters on TNFα or IL-6 production (Table 2).
- The effect of enzymes (such as MMP9) or transcription factors (such as Nrf2) is based not only on their gene or protein expression, but also on their activity. To be able to make a statement about the role of these factors, you also need the corresponding activity data. Please insert them.
Response: We acknowledge the reviewer’s comment and have included a sentence in Section 4.1 (Page 11, Line 301-302) to note the absence of MMP-9 and Nrf2 activity data as a potential limitation. “While MMP-9 protein concentration and Nrf2 mRNA expression was reported for macrophage lysates, further studies could correlate these end-points with their activity.”
Reviewer 2 Report
Endotoxin tolerance in abdominal aortic aneurysm macrophages, in vitro: a case-control study. Bye Meital et al.
Line 152: “PGE2 metabolite levels (F; AAA n=35, control n=34). “ How come there are 35 AAA samples here, while in the table there are not more than 19 AAA patients. Although counting the patients from the confocal it would add up to 33?
Figure 2. Line 180: “Transforming growth factor-β (TGF-β; E) concentrations tended to be higher in AAA macrophage supernatants, although this did not achieve significance. Catalase activity (H) and expression (I) and glutathione peroxidase (GPx) activity (J) and expression (K) were similar to the control cohort.” How many samples per assay were used for these biomarkers? It is not stated and since the number of patient derived macrophages varies for every biomarker assayed it is unclear.
For both remarks: Why are the numbers of samples used for each assay so different. It would have been more robust if all assays would have been performed with all samples.
Line 222: “Levels of phosphorylated NF-lB p65protein subunit”. Should be a kappa, I presume for NF-kB.
Figure 4. Line 214: “AAA macrophage lysate TLR4 mRNA (I) and protein levels (J) were similar to control. TLR4 mRNA and protein levels were normalized to GAPDH.” How many samples for mRNA and protein per group??
There is a nice review published recently (J Pathol 2020; 250: 705–714), which indicates good markers for different macrophages subsets. Can’t the authors use any of these markers to demonstrate what type of macrophages/response it is by determining a set of markers by qPCR for the whole cohort of samples? Atherosclerotic lesion cholesterol-loaded macrophages are also known to are less inflammatory. In that light it is known that cholesterol accumulation in the membranes determines lipid rafts, thus TLR4 signalling (J Immunol 2011; 187:1529-1535). It seems there is a cholesterol related phenotype? Are the blood cholesterol levels of all these patients measured and different? The statin use suggests cholesterol differences and the statins may explain the phenotype. Please elaborate.
Author Response
Reviewer #2
Endotoxin tolerance in abdominal aortic aneurysm macrophages, in vitro: a case-control study. Bye Meital et al.
1. Line 152: “PGE2 metabolite levels (F; AAA n=35, control n=34). “ How come there are 35 AAA samples here, while in the table there are not more than 19 AAA patients. Although counting the patients from the confocal it would add up to 33?
Response: Thank you for identifying this anomaly. The cause of the anomaly in AAA numbers was that we have included bloods from 18 patients in the biomarker cohort, 12 patients in the confocal cohort and an additional 5 patients who were not in either cohort and therefore also not included in Table 1. To rectify the problem, we have re-analyzed the data after removing the additional group of patients. The interpretation of findings is unchanged. The revised data includes n=30 patients with AAA (Page 4, Line 122), mean±SEM=15.33±1.89 pg/mL and P=0.014. Line 122 has been corrected and Figure 1f has been updated with the revised data.
2. Figure 2. Line 180: “Transforming growth factor-β (TGF-β; E) concentrations tended to be higher in AAA macrophage supernatants, although this did not achieve significance. Catalase activity (H) and expression (I) and glutathione peroxidase (GPx) activity (J) and expression (K) were similar to the control cohort.” How many samples per assay were used for these biomarkers? It is not stated and since the number of patient derived macrophages varies for every biomarker assayed it is unclear.
Response: N values have now been included for the results that were not significantly different (Page 7, Lines 151-153):
TGF-β, control, n=14; AAA, n=10
Catalase activity, control, n=7; AAA, n=7
GPx activity, control, n=8; AAA, n=8
Catalase Western blot, control, n=4; AAA, n=6
GPx Western blot, control, n=4; AAA, n=6
3. For both remarks: Why are the numbers of samples used for each assay so different. It would have been more robust if all assays would have been performed with all samples.
Response: We agree with the reviewer. However, a finite number of monocytes were obtained from blood samples collected from each participant and this resulted in a limited amount of cell culture supernatant that could be used to run ELISAs. For this reason, not all assays could be run using all samples.
4. Line 222: “Levels of phosphorylated NF-lB p65protein subunit”. Should be a kappa, I presume for NF-kB.
Response: We have corrected the formatting issue by replacing “I” with “κ”
5. Figure 4. Line 214: “AAA macrophage lysate TLR4 mRNA (I) and protein levels (J) were similar to control. TLR4 mRNA and protein levels were normalized to GAPDH.” How many samples for mRNA and protein per group??
Response: N values have now been included for these results (Page 9, Lines 188-189):
TLR4 mRNA, AAA, n=9; Control, n=8
TLR4 protein, AAA, n=7; Control, n=5
6. There is a nice review published recently (J Pathol 2020; 250: 705–714), which indicates good markers for different macrophages subsets. Can’t the authors use any of these markers to demonstrate what type of macrophages/response it is by determining a set of markers by qPCR for the whole cohort of samples?
Response: The experimental design ensured that the macrophages (obtained from circulating monocytes), were polarized to an M1 phenotype. On Day 7 of cell culture, cells were incubated with 20 ng/mL interferon-γ and 0.1 μg/mL of E.coli-derived lipopolysaccharide (LPS). Use of the LPS/IFN- γ combination to polarise monocyte-derived macrophages toward and M1 phenotype is based on previously published monocyte/macrophage studies (Jaguin et al., 2013; Ploeger et al., 2013; Matinez and Gordon, 2014; Tedesco et al., 2015). This study therefore highlighted that responses of pro-inflammatory M1 macrophages were blunted in AAA.
Jaguin, M. et al., Polarization profiles of human M-CSF-generated macrophages from GM-CSF and M-CSF origin. Cellular Immunology, 2013;281:51-61.
Ploeger, D. et al., Cell plasticity in wound healing: paracrine factors of M1/M2 polarized macrophages influence the phenotypical state of dermal fibroblasts. Cell Communication Signaling, 2013;11:29.
Matinez, F.O. and Gordon, S. The M1 and M2 paradigm of macrophage activation: time for reassessment. F1000 Prime Reports, 2014;6:13.
Tedesco, S. et al., Phenotypic activation and pharmacological outcomes of spontaneously differentiated human monocyte-derived macrophages. Immunobiology, 2015;220:545-554.
7. Atherosclerotic lesion cholesterol-loaded macrophages are also known to are less inflammatory. In that light it is known that cholesterol accumulation in the membranes determines lipid rafts, thus TLR4 signalling (J Immunol 2011; 187:1529-1535). It seems there is a cholesterol related phenotype? Are the blood cholesterol levels of all these patients measured and different? The statin use suggests cholesterol differences and the statins may explain the phenotype. Please elaborate.
Response: Although cholesterol-loading of macrophages may lead to a suppressed inflammatory state in atherosclerosis, many of the patients within the AAA cohort (74%) were being administered statins. The serum concentration of cholesterol was lower in the AAA patient group compared to control participants (AAA, 3.92±0.23 mmol/L, n=32; Control, 4.84±0.16 mmol/L, n=25; P=0.0015), consistent with the high use of statins. Since serum cholesterol concentration was lower in our patient group, it is unlikely that this factor would contribute to suppression of macrophage activity specifically caused by cholesterol loading. We also provided Mantel-Haenszel analysis of potential confounding drug therapies. The findings indicated that concentrations of TNF-α and IL-6 in macrophage supernatants were not different for patients with AAA who were administered statins (TNF-α,17.22±4.62 ng/mL; IL-6, 1.30±0.37 ng/mL) and patients who were not administered statins (TNF-α, 11.96±3.06 ng/mL; IL-6, 1.58±0.59 ng/mL). The Mantel-Haenszel analysis (Table 2) did not reveal a confounding influence of statins (TNF-α, P=0.28; IL-6, P=0.96) or hyperlipidemia (TNF-α, P=0.46; IL-6, P=0.73).
Reviewer 3 Report
Manuscript ID: antioxidants-905645
Title: Endotoxin tolerance in abdominal aortic aneurysm macrophages, in vitro: a case-control study
Authors: Lara T. Meital et al.
General comments; This study described an inflamed AAA environment allow macrophages to reduce their capacity to produce pro-inflammatory mediators that enhance the immune response. This study only provided accumulative evidence to the knowledge already known. However, the manuscript is well written with carefully executed experimental designs.
Specific comments;
The authors should explain why the study only choose male candidates not females.
In each result section, the authors need to explain the rational of each marker measured, e.g. why measure 8-isoprostane and TNF-a etc.
In LPS stimulation, why the authors measured MMP-9 etc. With rational, the readers would understand the importance of each biomarker related to the experimental manipulation.
Why the authors measured Nrf2?
Author Response
Reviewer #3
Manuscript ID: antioxidants-905645
Title: Endotoxin tolerance in abdominal aortic aneurysm macrophages, in vitro: a case-control study
Authors: Lara T. Meital et al.
General comments; This study described an inflamed AAA environment allow macrophages to reduce their capacity to produce pro-inflammatory mediators that enhance the immune response. This study only provided accumulative evidence to the knowledge already known. However, the manuscript is well written with carefully executed experimental designs.
Specific comments;
- The authors should explain why the study only choose male candidates not females.
Response: The risk of AAA is higher in men than in women (odds ratio, 3.24; Summers et al., 2020). Considerable evidence exists for gender differences in morphology and hemodynamic responses in AAA. Morphologically, women have a smaller volume of intraluminal thrombus than men (Gao et al., 2020). Hemodynamic differences include higher peak pressure and higher wall shear stress at the lower anterior area of the AAA, and lower oscillatory shear stress index for women compared to men (Gao et al., 2020). For patients with AAA, the risk of rupture is significantly greater for women than for men (Brown et al., 1999).
In this study, we isolated monocytes from whole blood and allowed spontaneous differentiation to macrophages in culture. Matrix metalloproteinase MMP-9 and cytokine levels were examined in the cells. mRNA expression of MMP-9 in non-thrombus-covered aneurysm wall has been reported to be significantly higher in women than in men with AAA (Villard et al., 2014). In an elastase-perfusion rat model of AAA, plasma concentrations of cytokines and chemokines (bone morphogenic protein, interleukins 1, 2, 3, 5, 7, 11 and 12; transforming growth factor β1 and β2; and vascular endothelial growth factors 1 and 2), were significantly different between male and female rats (Sinha et al., 2006).
Male only patients were included in this study to avoid a potential confounding influence of gender on measurements of cytokines and MMP-9. Further studies would be required to determine whether tolerance also exists in women with AAA. This has now been acknowledged as a future investigation within the limitations section (Page 11, Lines 304-305): “Further studies are also warranted to examine evidence of endotoxin tolerance in human AAA macrophages in situ, and to investigate whether similar responses are observed in macrophages obtained from female patients who have AAA.”
We have included the following sentence in the Method section, Page 2, Line 79-81: “Male participants only were recruited to this study as AAA is more prevalent in men than in women and sex differences exist with respect to AAA morphology and hemodynamics.”
Brown, L.C. and Powell, J.T. Risk factors for aneurysm rupture in patients kept under ultrasound surveillance. UK small aneurysm trial participants. Ann. Surg., 1999;230:289-296.
Gao, Z. et al., Gender differences of morphological and hemodynamic characteristics of abdominal aortic aneurysm. Biology of Sex Differences, 2020;11:41.
Sinha, I. et al., Female gender attenuates cytokine and chemokine expression and leukocyte recruitment in experimental rodent abdominal aortic aneurysms. Annals New York Academy Sciences, 2006;1085:367-379.
Summers, K. et al., Evaluating the prevalence of abdominal aortic aneurysms in the United States through a national screening database. Journal of Vascular Surgery, 2020, S0741-5214.
Villard, C. et al., Differences in elastin and elastolytic enzymes between men and women with abdominal aortic aneurysm. Aorta, 2014;2:179-185.
- In each result section, the authors need to explain the rational of each marker measured, e.g. why measure 8-isoprostane and TNF-a etc. In LPS stimulation, why the authors measured MMP-9 etc. With rational, the readers would understand the importance of each biomarker related to the experimental manipulation. Why the authors measured Nrf2?
Response: This information already exists in the Abstract and in the Method section of the manuscript. In the Abstract (Page 1, Line 30-31), it is stated: “LPS-stimulated production of 8-isoprostane, a biomarker of oxidative stress, was also markedly lower in AAA compared to control participants.” Page 1, Lines 26-27, state: “To establish oxidative stress status, free 8-isoprostane levels were measured in BHT-preserved plasma samples and monocyte-derived macrophage supernatants, as described previously [14].”
On Page 10, Lines 230-232, we stated: “The abrogated TNF- and IL-6 response and down-regulation of MMP-9 identified in AAA macrophages is characteristic of endotoxin tolerance as described by evidence from human studies [33,34].” MMP-9 was therefore selected as a biomarker of endotoxin tolerance.
On Page 10, Lines 255-257, we stated: “We measured Nrf2 mRNA levels in macrophages from AAA patients and control participants in light of recent evidence suggesting this transcription factor negatively regulates macrophage responses to LPS [46].” On Page 10, Lines 259-261, we stated: “As Nrf2 is reported to inhibit pro-inflammatory cytokine gene expression in M1 macrophages through direct binding to DNA, it is possible that this transcription factor represents a tolerizing mechanism in AAA.”
Round 2
Reviewer 1 Report
No further comments!
Reviewer 2 Report
I have nothing further